# Lipoprotein(a) and Low-Molecular-Weight Apo(a) Phenotype as Determinants of New Cardiovascular Events in Patients with Premature Coronary Heart Disease

**DOI:** 10.3390/diseases11040145

**Published:** 2023-10-18

**Authors:** Olga I. Afanasieva, Alexandra V. Tyurina, Marat V. Ezhov, Oxana A. Razova, Elena A. Klesareva, Sergei N. Pokrovsky

**Affiliations:** 1Institute of Experimental Cardiology, National Medical Research Center of Cardiology Named after Academician E.I. Chazov, Ministry of Health of the Russian Federation, Academician Chazov str., 15a, 121552 Moscow, Russia; afanasieva.cardio@yandex.ru (O.I.A.); razova1@yandex.ru (O.A.R.); hea@mail.ru (E.A.K.); dr.pokrovsky@mail.ru (S.N.P.); 2A.L. Myasnikov Institute of Clinical Cardiology, National Medical Research Center of Cardiology Named after Academician E.I. Chazov, Ministry of Health of the Russian Federation, Academician Chazov str., 15a, 121552 Moscow, Russia; alex.tyurina.cardio@yandex.ru

**Keywords:** lipoprotein(a), apo(a) phenotype, PCSK9, PCSK9-Lp(a) complex, myocardial infarction, cardiovascular events, premature coronary heart disease

## Abstract

Background. Lipoprotein(a) (Lp(a)) is a genetic risk factor of atherosclerotic cardiovascular diseases (ASCVDs). Proprotein convertase subtilisin/kexin type 9 (PCSK9) is related to vascular inflammation and detected in atherosclerotic plaques. A temporary increase in the circulating concentration of PCSK9 and Lp(a) was shown in patients with myocardial infarction (MI). The aim of this study was to evaluate the role of the apo(a) phenotype and the Lp(a) concentration as well as its complex with PCSK9 in the development of cardiac events and MI in patients with a premature manifestation of coronary heart disease (CHD). Methods. In a prospective study with retrospective data collection, we included 116 patients with premature CHD who were followed for a median of 14 years. The medical history and information on cardiovascular events after an initial exam as well as data on the levels of lipids, Lp(a), PCSK9, PCSK9-Lp(a) complex, and apo(a) phenotype were obtained. Results. The patients were divided into two groups depending on the presence of a low- (LMW, *n* = 52) or high-molecular weight (HMW, *n* = 64) apo(a) phenotype. LMW apo(a) phenotype (odds ratio 2.3 (1.1 to 4.8), *p* = 0.03), but not elevated Lp(a) (1.9 (0.8–4.6), *p* = 0.13), was an independent predictor for the development of MI after adjustment for sex, age of CHD debut, initial lipids levels, and lipid-lowering treatment. The apo(a) phenotype also determined the relationship between Lp(a) and PCSK9 concentrations. The level of the PCSK9-Lp(a) complex was higher in LMW apo(a) patients. Conclusion. The LMW apo(a) phenotype is a risk factor for non-fatal MI in a long-term prospective follow-up of patients with premature CHD, and this link could be mediated via PCSK9.

## 1. Introduction

Lipoprotein (a) (Lp(a)) is a genetically determined risk factor for the development of atherosclerotic cardiovascular diseases (ASCVDs) and is associated with cardiovascular events even in patients receiving optimal lipid-lowering therapy [1,2]. Lp(a) consists of a particle similar to low-density lipoprotein (LDL), where apolipoprotein B100 (apoB) is bound by a single disulfide bond with apolipoprotein(a) (apo(a)). Apo(a) is highly glycosylated and possesses a kringle structure, a high degree of homology with plasminogen, and polymorphisms. The *LPA* gene sequence explains up to 90% of the variation in Lp(a) concentration; 40–70% of these variations are determined by the size of the apo(a) isoforms. The molecular weight of apo(a) varies in the range from 300 to 800 kDa and is inversely related to Lp(a) concentration in human plasma due to the lower rate of apo(a) synthesis with a higher-molecular-weight (HMW) apo(a). However, the relationship between apo(a) isoform size and Lp(a) concentration is not absolute and is modified by various polymorphisms, resulting in a wide variability in the Lp(a) concentration in individuals with the same isoform [2].

The contribution of apo(a) isoforms to Lp(a) atherothrombogenesis has been discussed for a long time, but the question remains open. This is mainly due to the variations in the method for determining Lp(a), the difference in the methods used for phenotyping apo(a), and the patient groups. We have previously shown that the low-molecular-weight (LMW) apo(a) phenotype is associated with severe coronary atherosclerosis and myocardial infarction (MI) in patients with Lp(a) levels above 50 mg/dL, especially in those younger than 50 years [3]. The significance of LMW apo(a) as a risk factor for MI could not be determined retrospectively. The association of pro-inflammatory and pro-atherogenic oxidized phospholipids (OxPLs) with allele-specific Lp(a) levels in healthy adults and children [4] suggests that the LMW apo(a) phenotype may be an additional risk factor for ASCVDs regardless of the concentration of Lp(a).

It has recently been shown that the use of alirocumab, which causes a slight (up to 25%) decrease in Lp(a) level, is associated with a reduced risk of cardiovascular outcomes after acute coronary syndrome (ACS) despite a greater decrease in LDL cholesterol concentration [5]. The mechanism of how proprotein convertase subtilisin/kexin type 9 (PCSK9) inhibition decreases Lp(a) concentration is still unclear [6]. However, the size of the apo(a) isoform affects the degree of Lp(a) level reduction under the action of monoclonal antibodies to PCSK9 [7]. Previously, we showed that, in patients with familial hypercholesterolemia (FH), the presence of the LMW, but not the HMW, apo(a) phenotype, determined the relationship between Lp(a) and PCSK9 concentrations [8]. The existence of the PCSK9-Lp(a) complex circulating in human blood plasma has also been described in patients with elevated Lp(a) levels [9], FH, and healthy subjects [8,10,11]. Increased PCSK9 expression by ischemic cardiomyocytes in acute MI [12] and other effects of PCSK9 on immune and endothelial cells [13] could be responsible for the high thrombogenic properties of Lp(a). Immune blood cells, including all monocytes, are involved in the process of inflammation and atherosclerosis, and the ratio of lymphocytes to monocytes (LMR), reflecting chronic inflammation, is associated with the prognosis of patients with ASCVDs [14]. 

Considering that Lp(a) is a genetically determined risk factor, its role in the secondary prevention of ASCVDs is an emergent issue. More attention is being paid to the possibility of using PCSK9 inhibitors for the treatment of ASCVDs and MI [15]. The aim of this study was to evaluate the role of the LMW apo(a) phenotype and the Lp(a) concentration as well as its complex with PCSK9 in the development of cardiac events and MI in patients with premature CHD.

## 2. Materials and Methods

A prospective study with retrospective data collection included 116 patients with premature CHD (men before 55 years, women before 60 years) [16] who were followed up from 1994 to 2021 (Figure A1). The CHD was confirmed by the presence of stenosis in one or more coronary arteries ≥50% with primary coronary angiography (CAG) or myocardial ischemia at stress testing or documented myocardial revascularization or MI. A medical history was collected in all patients; the lipid profile and Lp(a) concentration were measured, and the apo(a) phenotype was determined. Data on adverse cardiovascular events, including non-fatal MI, coronary artery bypass grafting (CABG) surgery, and unstable angina requiring hospitalization, during the follow-up period after CHD manifestation were obtained from medical records. The median observation period and interquartile intervals were 14 [11,16] years. The exclusion criteria were chronic renal or liver failure, malignant neoplasms and autoimmune diseases, human immunodeficiency virus, hepatitis B and C, mental disorders, and alcohol abuse. We also did not include those patients who received therapy affecting the concentration of Lp(a) (PCSK9 inhibitors, lipoprotein apheresis) and had a concentration of Lp(a) below 4 mg/dL, which did not allow us to determine the phenotype of apo(a).

Total cholesterol (TC), triglycerides (TGs), high-density lipoprotein cholesterol (HDL-C), and Lp(a) levels and apo(a) phenotypes were determined in all patients. The concentrations of PCSK9 and the PCSK9-Lp(a) circulating complex were measured in all patients at a follow-up visit. The LDL-C concentration was calculated according to the Martin–Hopkins equation [17]. NonHDL-C and cholesterol remnant lipoproteins (RLP-C) were calculated as follows: nonHDL-C = TC − HDL-C and RLP-C = TC − LDL-C − HDL-C [18]. The level of LDL-C corrected (LDL-Ccorr) was estimated with the Dahlen modification of the Friedewald formula: LDL-Ccorr = LDL-C − 0.3 × Lp(a)/38.7 [19]. The Lp(a) concentration was measured using an intralaboratory enzyme-linked immunosorbent assay as previously reported [20]. Levels of PCSK9 and PCSK9-Lp(a) were determined through an ELISA with a commercial kit (R&D Systems, Minneapolis, MN, USA) and intralaboratory test with monoclonal antibodies against PCSK9 and polyclonal antibodies against Lp(a), respectively.

Apo(a) phenotyping was performed through Western blotting as previously described [20]. The apo(a) phenotypes were divided according to apoB mobility [21]. The HMW apo(a) phenotype group included patients with apo(a) major band mobility lower than that of apoB (more than 22 KIV2 repeats). In the LMW apo(a) phenotype group, the mobility of the apo(a) major band was greater than that of apoB-100 (up to 22 KIV2 repeats). All samples of serum were kept at 70 °C until use.

The Kolmogorov–Smirnov test was used to determine the normality of the distribution. The data are presented as mean ± standard deviation (SD) for those with normal distribution or as median and interquartile range or 95% confidential interval for those with abnormal distribution. The Spearman’s test was used for correlation analysis. We conducted logistic regression to determine the association of independent variables with cardiovascular events and MI. Survival analysis without non-fatal cardiac events was carried out through regression analysis of Kaplan–Meier and Cox. Statistical significance was considered at *p* < 0.05.

## 3. Results

All patients were divided into two groups depending on the presence of the LMW (*n* = 52) or the HMW (*n* = 64) apo(a) phenotype. Patients did not differ by main clinical characteristics, age of CHD manifestation, and lipid values (Table 1 and Table 2). There were more females in the LMW apo(a) phenotype group.

The concentration of Lp(a) was significantly higher in patients with LMW apo(a) phenotypes (Figure 1).

During the observation period, cardiovascular events developed in 71 patients (Table A1).

The risk of cardiovascular events was independent of apo(a) phenotype and Lp(a) concentration at a cutoff value of 30 mg/dL (Figure 2A,B). An Lp(a) level above 50 mg/dL was significantly associated with an increased risk of events by 67% (Figure 2C). The mean event-free survival time in patients with Lp(a) levels above 50 mg/dL was 5 years shorter than in patients with Lp(a) < 50 mg/dL (116 ± 15 months versus 180 ± 24 months, respectively).

The presence of the LMW apo(a) phenotype was associated with a 4-year-earlier development of MI compared to the HMW apo(a) phenotype: 226 ± 13 months versus 276 ± 15 months, respectively. The risk of MI was 2.4 times higher in the presence of the LMW apo(a) phenotype (Figure 3A) and twice higher in patients with an Lp(a) above 30 and 50 mg/dL, but this association did not reach statistical significance (Figure 3B,C).

The LMW apo(a) phenotype (HR = 2.29; 1.08–4.81, *p* = 0.03) was an independent predictor of earlier development of non-fatal MI, according to Cox regression analysis adjusted for sex, age at CHD manifestation, baseline LDL-C and HDL-C concentrations, and statin administration. When both Lp(a) concentration and apo(a) phenotype were simultaneously included in the model as independent variables, only the LMW apo(a) phenotype increased the risk of non-fatal MI: HR = 2.31; 1.06 to 5.00, *p* = 0.03. 

The Cox proportional risk model of non-fatal MI in subgroups of patients divided by the presence or absence of elevated Lp(a) level and LMW apo(a) phenotype showed that the presence of the LMW apo(a) phenotype, even at Lp(a) concentrations below 30 mg/dL, predicts the risk of non-fatal MI but not cardiac events (Figure 4).

The concentration of the circulating PCSK9-Lp(a) complex was significantly higher in patients with the LMW apo(a) phenotype than in patients with the HMW apo(a) phenotype, while the difference between the groups in terms of PCSK9 levels was not significant (Figure 5).

Correlation analysis revealed the relationship between the concentration of Lp(a) and PCSK9 in the whole group (*r* = 0.48, *p* = 0.0007) and in patients with the LMW apo(a) phenotype (*r* = 0.60, *p* = 0.003) but not the HMW apo(a) phenotype (*r* = 0.14, *p* = 0.50).

According to logistic regression analysis, an increase in PCSK9 concentration by 1 pg/mL was associated with an increase in the odds ratio of events by 1% (OR = 1.01; 1.00–1.02, *p* = 0.03) only in patients with the LMW apo(a) phenotype. PCSK9 levels above the median were associated with an increased odds ratio of cardiovascular events (OR = 6.67; 0.99–45.04, *p* = 0.05) only in patients with the LMW apo(a) phenotype. 

In the whole cohort, we did not find a significant association between the concentration of PCSK9 or its circulating complex PCSK9-Lp(a) and risk of MI during the observation period. However, in the presence of a PCSK9-Lp(a) complex level above the median (91.8 lab units), MI occurred on average 17 years earlier in the LMW apo(a) phenotype patients compared to patients with the HMW apo(a) phenotype: 229 ± 36 months vs. 439 ± 50 months, respectively, *p* = 0.08 (Figure A2). In patients with a PCSK9-Lp(a) level < 91.8 lab units, this difference was non-significant: 290 ± 39 months vs. 302 ± 20 months (*p* = 0.74).

## 4. Discussion

CHD and subsequent cardiovascular events in subjects under the age of 55 years represent a serious medical and social problem [22,23]. The underestimation of traditional risk factors, including smoking and low physical activity, as well as limited access to healthy nutrition and medical care can cause up to 31% of CHD cases [24]. Individuals with premature CHD have a high proportion of modifiable cardiovascular risk factors. In studies determining the risk profile in young patients with CHD, the presence of smoking, hypertension, and a family history of CVD was most common [25]. Type 2 diabetes occurs in 14.7 to 38.1% of patients with early CVD [26], which is significantly lower than in patients with a later CHD debut [27]. Young patients with CHD have a higher body mass index and are more likely to be obese compared to controls of the same age and gender [28]. In our study, the analyzed groups did not differ in the frequency of classical risk factors for atherosclerosis.

Lp(a) is one of the most important genetically determined risk factors for the development of ASCVDs [29,30,31]. The number of repeats of the region encoding the copy number of one of the apo(a) domains (kringle IV type-2 repeat, KIV_2_) generates > 40 isoforms of the apolipoprotein(a) protein and determines the median concentrations of Lp(a). In the population, LMW apo(a) isoforms are associated with approximately higher average Lp(a) concentrations than HMW apo(a) isoforms [2]. However, this association may be confounded by the contribution of single-nucleotide polymorphisms and others genetic factors [32,33]. 

For the first time, it was shown in a long-term prospective observational study that the LMW apo(a) phenotype is the most important risk factor for the development of MI in patients with early-onset CHD. In contrast, for other cardiovascular events, elevated an Lp(a) level was more significant. We have also shown for the first time a possible association between apo(a) phenotype, PCSK9-Lp(a) complex levels, and adverse CHD course. This may be due to the greater “pro-inflammatory” activity of Lp(a) with the LMW apo(a) phenotype and due to the presence of both OxPL and PCSK9 in the Lp(a) particles.

We observed that Lp(a) concentrations ≥ 50 mg/dL increased the risk of cardiovascular events by 67% in patients with premature CHD (up to 55 in men and 60 in women). Our study considers a long observation period from the point of CHD manifestation. The Lp(a) level in these patients was significantly higher (33 [11; 81] mg/dL) than in patients without stenosing atherosclerosis (12 [8,9,10,11,12,13,14,15,16,17] mg/dL, *p* < 0.001) [30,34], and in the European population 19.6 [7.6–74.8] nmol/L (about 8.2 mg/dL) [35]. The Mendelian randomization study included 9015 patients, and 8629 controls demonstrated that both apo(a) size and Lp(a) concentration were independent risk factors for MI [31,36]. 

We have shown that, in patients with early CHD manifestation, elevated levels of Lp(a), but not LMW apo(a) phenotype, increased the risk of cardiovascular events during long-term follow-up despite ongoing statin therapy and correction of modifiable risk factors. In support of this observation, the risk of recurrent events associated with elevated Lp(a) levels was the highest in 2527 young CHD subjects from the Copenhagen General Population Study. This may be explained by the absence in young subjects of some risk factors that accumulate with age and mask a causal relationship with Lp(a) [1].

Exposure to such risk factors as hypertension, hyperglycemia, and smoking leads to the activation of vascular smooth muscle and endothelial cells, predominantly in areas of impaired laminar blood flow. The sequential increase and activation of adhesion molecules, in particular, ICAM-1, -2; VCAM-1; and E- and P-selectins on endothelial surface, leads to attraction and infiltration of immune cells, mainly monocytes, into the intimal layer, promoting low specific inflammation in the vascular wall [37,38]. Expression of the scavenger receptors CD36, LOX-1, and SR-A by activated macrophages leads to even greater cholesterol accumulation, assembly, and activation of the inflammasome, which completes the vicious circle by activating the immune response cascade [39]. Endothelial cell culture and transgenic mouse experiments have recently shown that Lp(a) can activate endothelium by affecting glycolysis, leading to increased monocyte adhesion and migration. Reducing Lp(a) levels with antisense oligonucleotides targeting apo(a) synthesis diminished glycolytic and inflammatory responses in ex vivo experiments in a human aortic endothelial cell culture, demonstrating a direct effect of Lp(a) on the endothelium [40].

An Lp(a) concentration more than 20 mg/dL was associated with increasing risk of heart failure due to MI in the Copenhagen City Heart Study and the Copenhagen General Population Study [41]. According to coronary computer tomography angiography findings, Lp(a) concentrations between 9 and 20 mg/dL were associated with the presence of uncalcified and vulnerable coronary plaques in asymptomatic patients [42]. One of the mechanisms underlying most MIs is vulnerable plaques with a large lipid core and a thin fibrous cap, which are prone to rupture with subsequent atherothrombosis [43,44,45]. Recently, a growing body of evidence has accumulated that immunity and inflammation contribute significantly to the development of MI, demonstrating that plaque stability is mainly dependent on the amount of inflammatory cell infiltration [46].

The potential role of Lp(a) as an activator of immune system cells is actively debated [47]. In particular, monocytes isolated from the blood of individuals with elevated Lp(a) levels showed increased secretion of proinflammatory cytokines and decreased secretion of anti-inflammatory cytokines [48]. Intimal macrophages activated by atherogenic lipoproteins produce not only proinflammatory cytokines such as TNF-α and IL-6 but also matrix metalloproteinases (MMPs). The involvement of matrix metalloproteinases, in particular matrix metalloproteinase-9 (MMP-9), in plaque instability and rupture has been discussed for decades and is still relevant [49]. The level of MMP-9 was the only independent predictor of atherosclerotic plaque rupture, and patients with diagnosed atherosclerotic plaque rupture had significantly higher levels of MMP-9 than patients without rupture [50]. In addition, the level of matrix metalloproteinases correlated with the absolute area of the necrotic core [51]. In our previous study of patients with stable CHD, an elevated Lp(a) concentration was strongly correlated with MMP-9, MMP-7, and the size of the necrotic core [52]. 

In vitro experiments showed that apo(a) was able to increase the production of reactive oxygen species and MMP-9 by monocytes stimulated by type I collagen. The degree of stimulation was inversely proportional to the molecular weight of apo(a) and was maximal when stimulated with the LMW apo(a) phenotype [53]. 

According to a number of studies, Lp(a) is the carrier of a major part of the total pool of OxPL [54]. Such oxidation-specific epitopes on apo(a) [55] are one of the key factors involved in the immune response and inflammation and determine the high “atherothrombogenicity” of Lp(a) [48,56]. In Europeans, levels of Lp(a)-associated OxPL were associated with allele-specific Lp(a) levels carried on smaller apo(a) sizes [4]. A possible explanation for this relationship between OxPL and apo(a) could be the ability of the LMW apo(a) phenotype to circulate longer in the bloodstream [57], that, in turn, could be due to interactions with different types of receptors [58]. A decrease in the concentration of Lp(a) and OxPL by antisense oligonucleotides leads to a decrease in inflammation in the vessel wall [59].

The presence of apo(a) in atherosclerotic vascular lesions from the lipid band to plaque rupture, as well as its presence in the macrophage-enriched area, has been demonstrated in morphological and immunohistochemical studies [60,61]. Previously, our in vitro studies showed that THP-1 macrophage cells accumulate cholesterol much more actively when adding serum from patients with the LMW apo(a) phenotype, regardless of the content of other lipids and the initial concentration of Lp(a) [62]. Thus, one of the pathophysiological mechanisms of the greater “atherothrombogenicity” of the LMW apo(a) phenotype may be its ability to activate immune cells to a greater extent, thereby increasing and maintaining inflammation in the atherosclerotic plaque, making it more vulnerable. 

In our study, we confirmed our earlier observation [8] that the relationship between Lp(a) and PCSK9 concentrations is typical only for individuals with the LMW apo(a) phenotype. The PCSK9-Lp(a) complex content was higher in patients LMW apo(a) isoforms; this may also confirm a difference in Lp(a) metabolism with different apo(a) phenotypes. For the first time, we have shown that, only for patients with the LMW apo(a) phenotype, the level of PCSK9 is associated with cardiovascular events. PCSK9 has additional functions beyond its effect on lipid metabolism [63]. A temporary increase in the concentration of circulating PCSK9 as well as Lp(a) was shown in acute MI. At the same time, the Lp(a) level was suppressed by monoclonal antibodies to PCSK9; this suggests that PCSK9 and Lp(a) are related to MI, and PCSK9 inhibition could be the treatment option for acute MI [64]. We showed that the PCSK9 concentration was associated with an increased risk of events, which was confirmed by the results of a larger prospective study involving 2293 patients [63].

There is no doubt that an increased concentration of Lp(a) is a genetic risk factor for ASCVDs. According to the recent EAS Consensus statement, the concentration of Lp(a) should be determined at least once in a lifetime to assess the risk of cardiovascular diseases [29]. Cascade screening for Lp(a) is the most justified in the presence of a family history of premature ASCVDs or the identification of an index patient with Lp(a) excess [65]. The results of our study indicate the importance of both increased concentration of Lp(a) and LMW apo(a) phenotype in patients with established premature CHD. The importance of measuring Lp(a) concentration in routine clinical practice is extremely relevant in the era of development of new drugs for the specific reduction of elevated Lp(a) levels in plasma [66].

## 5. Study Limitations

The limitation of our study was the small number of patients that necessitates further investigation. The lack of initial data on the concentration of PCSK9 and Lp(a)-PCSK9 complex does not allow us to conclude that there is a direct relationship between these parameters and the development of MI or cardiovascular events.

## 6. Conclusions

We showed that the LMW apo(a) phenotype is a risk factor for non-fatal MI in a long-term prospective follow-up of patients with premature CHD. The presence of the LMW apo(a) phenotype determines the correlation of Lp(a) and PCSK9 concentrations and a higher level of circulating complex PCSK9-Lp(a). Furthermore, only in patients with the LMW apo(a) phenotype was an elevated PCSK9 concentration associated with cardiovascular events after CHD manifestation. 

## Figures and Tables

**Figure 1 diseases-11-00145-f001:**
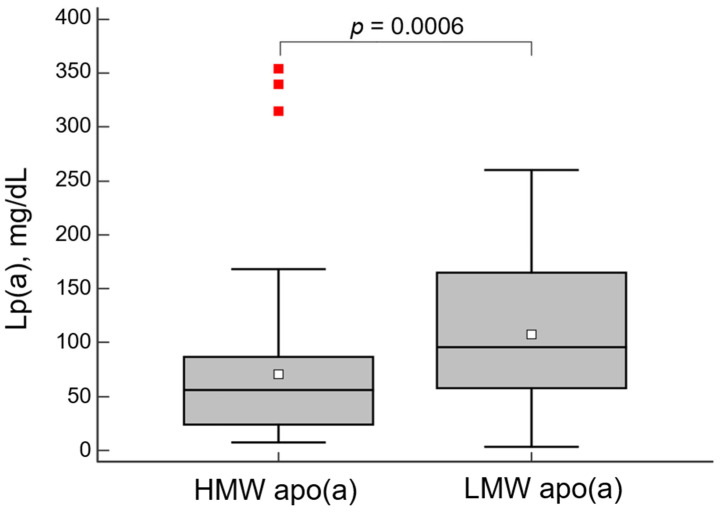
Lipoprotein(a) (Lp(a)) concentration in groups with high- (HMW) or low-molecular weight (LMW) apo(a) phenotype. The diagram is presented as a box and whiskers plot with the interquartile range. The horizontal line inside the box is the median, the unfilled square dot inside the box is the mean, whiskers—minimum and maximum, red squares—“far out value”—these are values higher than the upper outer fence (3rd quartile plus 3 times interquartile range).

**Figure 2 diseases-11-00145-f002:**
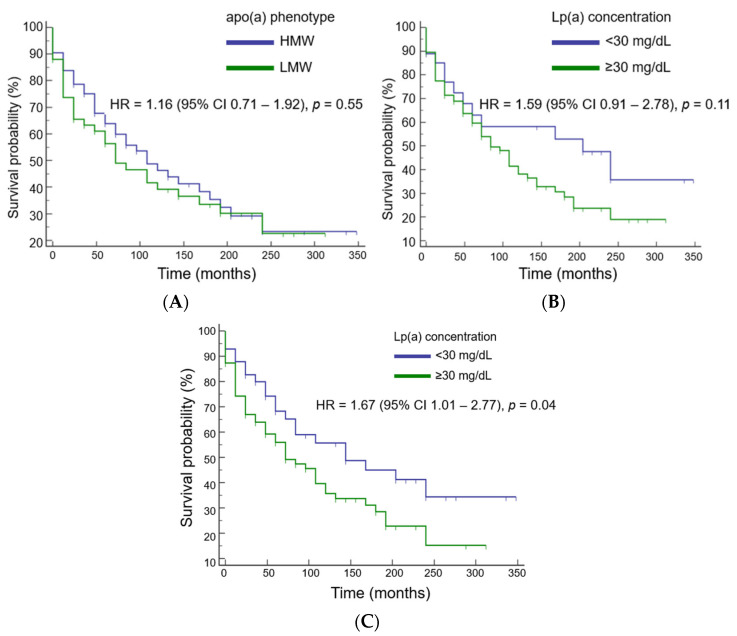
Cardiovascular event risk estimation dependent on apo(a) phenotype (**A**) and lipoprotein(a) level at a cutoff value of 30 mg/dL (**B**) or 50 md/dL (**C**).

**Figure 3 diseases-11-00145-f003:**
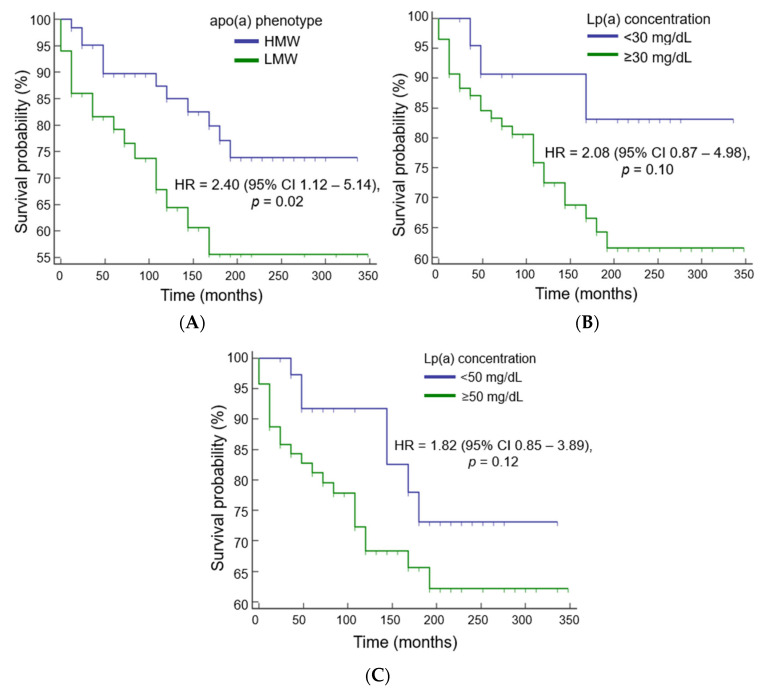
Myocardial infarction risk estimation dependent on apo(a) phenotype (**A**) and lipoprotein(a) cutoff value 30 mg/dL (**B**) or 50 md/dL (**C**).

**Figure 4 diseases-11-00145-f004:**
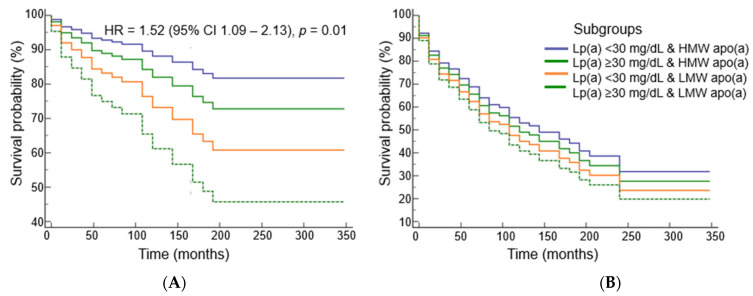
Survival probability without myocardial infarction (**A**) and cardiac events (**B**) dependent on lipoprotein(a) concentration and apo(a) phenotype (adjusted for sex, age of CHD manifestation, LDL-C_corr_, HDL-C levels, and statin intake).

**Figure 5 diseases-11-00145-f005:**
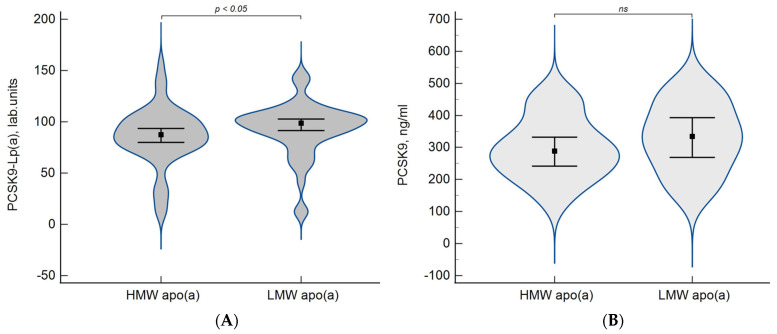
Levels of PCSK9-Lp(a) complex (**A**) and PCSK9 (**B**) depending on apo(a) phenotypes. Data are presented as a violin plot, squares indicate the median and whiskers indicate the 95% confidence intervals.

**Table 1 diseases-11-00145-t001:** Clinical characteristics of patients.

	HMW Apo(a) Phenotype *n* = 64	LMW Apo(a) Phenotype *n* = 52	*p*
Male gender	57 (89%)	35 (67%)	0.004
Age, years	59.3 ± 8.8	59.3 ± 9.1	0.69
Age at CHD manifestation, years	46.8 ± 7.6	45.7 ± 6.9	0.38
Hypertension	54 (83%)	46 (88%)	0.41
Type 2 diabetes	21 (32%)	15 (29%)	0.68
Smoking (current and past)	40 (61%)	32 (61%)	1.00
Statins	57 (89%)	50 (96%)	0.57
Antiaggregants	57 (89%)	48 (92%)	0.41

Data are presented as *n* (%) or mean ± standard deviation (SD).

**Table 2 diseases-11-00145-t002:** Lipid parameters at baseline and follow-up visits.

Lipids, mmol/L	HMW Apo(a) Phenotype *n* = 64	LMW Apo(a) Phenotype *n* = 52	*p*
CHD manifestation (visit 1)			
TC	5.65 [4.95; 6.75]	6.32 [5.19; 8.35]	0.06
TG	1.28 [0.99; 1.75]	1.60 [1.12; 2.30]	0.07
HDL-C	1.08 [0.94; 1.31]	1.11 [0.96; 1.32]	0.55
LDL-C	4.01 [3.24; 4.96]	4.06 [3.21; 6.18]	0.33
LDL-C_corr_	3.49 [2.69; 4.34]	3.36 [2.52; 5.37]	0.89
NonHDL-C	4.55 [3.75; 5.57]	4.88 [3.86; 7.10]	0.09
RLP-C	0.62 [0.49; 0.77]	0.69 [0.55; 0.90]	0.05
Follow-up (visit 2)			
TC	4.04 [3.38; 4.71]	4.19 [3.43; 4.74]	0.76
TG	1.35 [1.01; 1.75]	1.58 [1.06; 2.05]	0.16
HDL-C	1.07 [0.94; 1.32]	1.11 [0.95; 1.28]	0.65
LDL-C	2.34 [1.85; 2.84]	2.25 [1.71; 2.87]	0.60
LDL-C_corr_	1.76 [1.40; 2.38]	1.42 [0.96; 2.15]	0.04
NonHDL-C	2.93 [2.34; 3.48]	2.92 [2.35; 3.55]	0.96
RLP-C	0.56 [0.45; 0.68]	0.61 [0.51; 0.75]	0.11

TC—total cholesterol, TG—triglycerides, HDL-C—high-density lipoprotein cholesterol, LDL-C—low-density lipoprotein cholesterol, LDL-C_corr_—low-density lipoprotein cholesterol corrected for Lp(a) cholesterol, RLP-C—remnant lipoprotein cholesterol.

## Data Availability

The data presented in this study are available on request from the corresponding author.

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
