# Peer review of "Lipoprotein(a) and Low-Molecular-Weight Apo(a) Phenotype as Determinants of New Cardiovascular Events in Patients with Premature Coronary Heart Disease"

_diseases, 2023, doi:10.3390/diseases11040145_

Round 1

Reviewer 1 Report

Manuscript title: Lipoprotein(a) and low-molecular-weight apo(a) phenotype as determinants of new cardiovascular events in patients with premature coronary heart disease

Afanasieva and colleagues found that LMW apo(a) but not elevated Lp(a) was an independent predictor of development of MI after adjustment for sex, age of CHD debut, initial lipids levels and lipid-lowing treatment in a prospective study with retrospective data collection. The LMW apo(a) phenotype is a risk factor for non-fatal MI in a long-term prospective follow-up of patients with premature manifestation of CHD and this link could be mediated via PCSK9.

Major comments:

1. Based on the research published on Lp(a), apo(a) and PCSK9, the novelty of this paper is very low.

Minor comments:

1. Figure 1, please explain the red squares above the column in HMW apo(a).

2. Based on the text and figure legend, I think Figure 2 (B) should be 50 mg/dL. Please correct.

3. Figure 2 and figure 3, please arrange the order of the graphs (A, B and C) to match the order in the text.

Reviewer 2 Report

In this paper, the authors investigated genetic factor of premature coronary heart disease which is an interesting topic. Overall, it is well structured. Some improvements are still essential to meet the publishable standards.

1. Some previous studies (Refer: 10.4103/JCPC.JCPC_22_18) deserve more attention. I suggest add a paragraph to systematically review the existing studies and summarize the state-of-the-art research gaps.

2. It should be mentioned that premature coronary heart disease is defined differently in research. For example, male ≤35 years and females ≤40 years in 10.4103/JCPC.JCPC_22_18, before age 65 years in 10.1001/jamacardio.2020.1458, and before 50 years in 10.1161/JAHA.120.017712.

3. In the discussion, it should be mentioned that premature coronary heart disease has complex pathological mechanisms. Hemodynamic factors and endothelial function in epicardial arteries, as well as coronary microcirculatory dysfunction are well established risk factors of general coronary artery disease (Refer: 10.1088/2057-1976/aa9a09, 10.1016/j.compbiomed.2022.105583). In addition, some modifiable cardiovascular risk factors and comorbidities (Refer: 10.1161/JAHA.120.017712) including socioeconomic status (Refer: 10.1001/jamacardio.2020.1458) have also been identified as relevant to premature coronary heart disease. The risks factors also differ between sexes (Refer: 10.1089/jwh.2022.0517). The role of genetic factor need to be evaluated in a multifactor context in future large-scale studies.

4. It is also recommended to discuss possible underlying pathological mechanisms.

Overall it is well written. The writing and format can be further improved.

Reviewer 3 Report

The paper is excellent. There are significant limitations like small numbers and others mentioned by the authors but still very good material in an up and coming field depending on the Lpa meds being studied . The authors could include a Writing Group Lpa Guideline paper ( Virani SS et al. Prog Cardiovasc Dis 2022;73: 32-40) as well as recent State of the Art ( Krittanawong C et al. PCVD 2023; 79: 28-36.)

Author Response

Dear Reviewer,

We appreciate your positive response very much and have added the last paragraph to the Discussion section and cited both references.

Round 2

Reviewer 1 Report

The authors have addressed my concerns. No further comments. Thanks.

Author Response

Dear Reviewer,

We thank you for all your valuable comments and suggestions.

Reviewer 2 Report

Thanks for the update. Some of my earlier comments have been addressed. However, the discussion still need to be extended. 

In addition, several issues need further attention.

1. The statistical analysis should be clarified. What is the test and standard/threshold for normality? 

2. A frequently discussed condition of logistic regression is that the follow-up period is short or at least similar among subjects. Does it really applicable here? 

3. Figure 5 is confusing. Why the bar ends at 80/200? The distribution is unclear. I suggest using violin plot for figures 1 and 5. 

The writing and format can be further improved.

Author Response

  1. The statistical analysis should be clarified. What is the test and standard/threshold for normality? 

Thank you for your comment. The Kolmogorov-Smirnov test was used to determine the normality of the distribution. We added this phrase to the Methods section.

  1. A frequently discussed condition of logistic regression is that the follow-up period is short or at least similar among subjects. Does it really applicable here? 

Thank you for your comment. The follow-up period was comparable between the groups, so we consider that the use of logistic regression is applicable here.

  1. Figure 5 is confusing. Why the bar ends at 80/200? The distribution is unclear. I suggest using violin plot for figures 1 and 5. 

Thank you for your suggestion. We have changed the diagrams in Figure 5 to a violin plot. Figure 1 was left as the box-and-whiskers plot, but we added an explanation to the legend. Although diagrams in the form of a violin plot are more informative than box-and-whiskers, in our opinion, they may be harder to understand for readers.

  1. The writing and format can be further improved.

Thank you for your suggestion. We have rechecked the whole text with a native speaker and changed many phrases accordingly.

Round 3

Reviewer 2 Report

Thanks for the update. My earlier comments have been largely addressed. Some details in statistical terms and format still need further improvement. 

1. In the caption of Figure 1 you mentioned "far out values" do you mean outliers?

2. In line 122, it is "non-normal distribution" or "abnormal distribution". Please double check the terms.

3. In figure 5, the letters are not in line with the subfigures. The median is often shown with quartiles. Please add.

The language is improved and readable. 

Author Response

We thank both reviewers and the Academic Editor for time and help with our paper. We do not agree about the lack of novelty and below provide our rebuttal.

In our long-term prospective observational study evaluating patients with premature coronary heart disease, it was shown for the first time that the low molecular weight apo(a) phenotype is the most important risk factor for the development of MI after the manifestation of coronary heart disease. On the contrary, for other cardiovascular events, such as coronary artery bypass grafting and unstable angina requiring hospitalization, elevated Lp(a) levels were more significant. Given that drugs aimed at reducing the concentration of Lp(a) are at various stages of clinical trials, understanding the role of apo(a) phenotypes, in our opinion, is one of the unmet issues for both clinicians and researchers.

This study included only patients who had a premature manifestation of coronary heart disease, as a result of which the concentration of Lp(a) in this cohort of patients was significantly higher than in the European population as a whole, the PROMIS study, as well as other previously conducted studies. A median follow-up of 14 years also distinguishes our study from others.

The results obtained in recent years suggest that Lp(a) may be an activator of cells of nonspecific immunity. One of the possible molecular mechanisms of high atherothrombogenicity of Lp(a) may be the presence in the blood of a high content of the circulating PCSK9-Lp(a) complex and its potential effect on immune cells. For the first time, the relationship between the apo(a) phenotype, the levels of the PCSK9-Lp(a) complex and the development of MI in patients with premature coronary heart disease has no analogues. The phenomenon of a higher level of PCSK9-Lp(a) in patients with a low molecular weight apo(a) phenotype, as well as the association of PCSK9 with cardiovascular events only in patients with a low molecular weight apo(a) phenotype have not been published before. The observations made in our small study may be extremely important in light of the relationship we have identified between the content of the PCSK9-Lp(a) complex and the number of blood monocytes in patients with atherosclerosis [Filatova AY, Afanasieva OI, Arefieva TI, et al. The Concentration of PCSK9-Lp(a) Complexes and the Level of Blood Monocytes in Males with Coronary Atherosclerosis. Journal of Personalized Medicine. 2023; 13(7):1077. https://doi.org/10.3390/jpm13071077].
